# Pathogenic Microglia Orchestrate Neurotoxic Properties of Eomes-Expressing Helper T Cells

**DOI:** 10.3390/cells12060868

**Published:** 2023-03-10

**Authors:** Chenyang Zhang, Ben Raveney, Fumio Takahashi, Tzu-wen Yeh, Hirohiko Hohjoh, Takashi Yamamura, Shinji Oki

**Affiliations:** 1Department of Immunology, National Institute of Neuroscience, NCNP, Tokyo 187-8502, Japan; 2Department of Molecular Immunology, Graduate School of Medical and Dental Sciences, Tokyo Medical and Dental University, Tokyo 113-8549, Japan; 3Department of Molecular Pharmacology, National Institute of Neuroscience, NCNP, Tokyo 187-8502, Japan

**Keywords:** microglia, neurodegeneration, progressive multiple sclerosis

## Abstract

In addition to disease-associated microglia (DAM), microglia with MHC-II and/or IFN-I signatures may form additional pathogenic subsets that are relevant to neurodegeneration. However, the significance of such MHC-II and IFN-I signatures remains elusive. We demonstrate here that these microglial subsets play intrinsic roles in orchestrating neurotoxic properties of neurotoxic Eomes^+^ Th cells under the neurodegeneration-associated phase of experimental autoimmune encephalomyelitis (EAE) that corresponds to progressive multiple sclerosis (MS). Microglia acquire IFN-signature after sensing ectopically expressed long interspersed nuclear element-1 (L1) gene. Furthermore, ORF1, an L1-encoded protein aberrantly expressed in the diseased central nervous system (CNS), stimulated Eomes^+^ Th cells after Trem2-dependent ingestion and presentation in MHC-II context by microglia. Interestingly, administration of an L1 inhibitor significantly ameliorated neurodegenerative symptoms of EAE concomitant with reduced accumulation of Eomes^+^ Th cells in the CNS. Collectively, our data highlight a critical contribution of new microglia subsets as a neuroinflammatory hub in immune-mediated neurodegeneration.

## 1. Introduction

Neurodegeneration is the status/process of neuronal cell death or the functional breakdown of neuronal cells as part of the progressive failure of the nervous system in neurodegenerative diseases [1,2]. Intriguingly, neurodegeneration not only occurs in typical neurodegenerative diseases such as amyotrophic lateral sclerosis (ALS) and Alzheimer’s disease (AD) but also in chronic neuroinflammatory diseases [3] including secondary progressive multiple sclerosis (SPMS). Genetic approaches have identified a number of target molecules related to protein misfolding/aggregation, impaired RNA metabolism, mitochondrial dysfunction, dysregulated epigenetics, etc. [4,5,6,7]. Despite much research, the authentic causes of neurodegeneration are still undetermined. Mechanisms discerned from both human and animal studies of neurodegeneration are at best speculative and retrospective, thus yielding no possible prospect of therapeutic intervention. Recently, glial cells have been revealed as an intrinsic component in neurodegeneration [8], leading to acceptance as a possible cause or trigger of neuronal cell death [9,10,11], thereby establishing a non-cell autonomous hypothesis of neurodegeneration.

Microglia are neuro glia cells that function as resident macrophages in the CNS and form more than 10% of brain cellularity. These cells are involved in the active defense of the central nervous system (CNS) under physiological conditions by responding to and eliminating exogenous insults [12]. In addition, microglia are essential for the maintenance of brain architecture by clearing damaged neurons and pruning synapses during brain development [13]. Due to the broad range of tasks linked to CNS homeostasis, microglia are extremely sensitive to pathological fluctuations in the CNS [14]. Recent single cell-RNA seq-based analyses point out that the context-dependent heterogeneity of microglia during neurodegeneration [15,16,17,18] and disease-associated microglia (DAM), a microglial subset with highly pathogenic properties, has attracted attention in a mouse model of AD [18]. In addition to DAM, two microglial subsets with high expression of MHC-II and IFN-I-induced genes are highlighted in neurodegeneration, and these microglia may precede the final differentiation of DAM and could help unravel the pathogenic characteristics of microglia [19,20]. However, the pathogenic features of such microglial subsets remain undetermined, as the enhanced IFN-I signature and upregulated MHC II expression in themselves do not have any explicable relevance to neurodegeneration.

SPMS presents with progressive symptoms of brain atrophy, higher brain dysfunction, gait disturbance, and cognitive disorder and develops in MS patients following a period of times with an acute type of disease, relapsing remitting MS (RRMS) [21,22]. Immune responses were not previously though to contribute to SPMS pathogenesis; however there is growing evidence of an important role for active immune functions in SPMS. We have previously demonstrated that a unique population of T helper cells expressing Eomes (Eomes^+^ Th cells) are expanded in the CNS in a mouse model of SPMS, late, chronic EAE [23]. Furthermore, Eomes^+^ Th cells are increased in the peripheral blood and cerebrospinal fluid (CSF) of SPMS patients where the level of these cells was associated with actively progressing disease and so acted as a biomarker to predict SPMS patients at risk of developing worsening disease [24]. Eomes^+^ Th cells were also found at high levels in the CNS of brain autopsy samples from SPMS [24]. We have demonstrated in mouse models of amyotrophic lateral sclerosis (ALS) and Alzheimer’s disease (AD) that Eomes^+^ Th cells infiltrate into the CNS during neurodegenerative disease where they secrete neurotoxic granzyme B after encountering ectopically expressed ORF1 antigen encoded by L1 retrotransposon [25]. Ectopic derepression of nucleic acids such as L1 may cause a massive induction of IFN-I expression, forming a foundation of chronic inflammation, but the potential pathogenic significance of IFN-I during neurodegeneration is not well characterized. Although IFN-I is associated with reduced conventional RRMS and its mouse model acute EAE [26], constitutive IFN-I production in the CNS causes inflammatory symptoms designated as interferonopathies represented by Aicardi–Goutières syndrome and related diseases [27]. Furthermore, prolonged microglial IFN-I production exacerbates EAE at its chronic phase [28], and microglial upregulation of MHC II expression is associated with many neuroinflammatory and neurodegenerative diseases [29]. In conventional EAE, microglia do not play a role as effective APCs [30], but their involvement in the chronic phase of the CNS diseases has not been extensively studied.

In this study, we demonstrated that primed microglia orchestrate late EAE via reinforcement of IFN-I and MHC II driving induction and functional activation of pathogenic Eomes^+^ Th cells. Furthermore, we identified L1 as a dominant prototypic antigen recognized by CNS Th cells, suggesting a previously unappreciated function of microglia in neurodegeneration.

## 2. Materials and Methods

### 2.1. Mice

All mice used were maintained in specific pathogen-free conditions in accordance with institutional guidelines. C57BL/6 mice were purchased from CLEA Japan, Inc. (Tokyo, Japan) C57BL/6 CD4-Cre NR4A2 ^fl/fl^ (NR4A2 cKO) mice were generated in house as described recently [23]. CX3CR1^CreERT2^, Trem2 ^fl/fl^, IFN-α/βR KO, CX3CR1-GFP, and Granzyme B KO mice were obtained from The Jackson Laboratory, USA. CX3CR1^CreERT2^ mice were crossed with Trem2 ^fl/fl^ mice to generate Tamoxifen-inducible, microglia-specific Trem2-deficient mice. Five-week CX3CR1^CreERT2^Trem2 ^fl/fl^ mice were i.p. injected with 1 mg of Tamoxifen dissolved in sunflower oil for 5 consecutive days. All animal experiments were approved by the Committee for Small Animal Research and Animal Welfare (National Center of Neurology and Psychiatry). All efforts were made to minimize animal suffering in clinical disease experiments.

### 2.2. EAE Induction and Scoring

Female mice, aged 6–8 weeks, were immunized subcutaneously with 50 mg MOG_35–55_ peptide (synthesized by Toray Research Center, Japan) containing 1 mg heat-killed Mycobacterium tuberculosis H37RA emulsified in complete Freund’s adjuvant (Difco, Franklin Lakes, NJ, USA). The mice were injected intraperitoneally (i.p.) with 100 ng of Pertussis toxin (List Biological Laboratories, Campbell, CA, USA) at the time of immunization and 2 days later. Neurological deficits were evaluated on a scale from 0 to 5 (0, no clinical signs; 0.5, tail weakness; 1, partial tail paralysis; 1.5, severe tail paralysis; 2, flaccid tail; 2.5, flaccid tail and hind limb weakness; 3, partial hind limb paralysis; 3.5, severe hind limb paralysis; 4, total hind limb paralysis; 4.5, hind and fore leg paralysis; 5, dead.).

### 2.3. Treatment of Animals

200 µg anti-mouse IFNAR or isotype control (all from BioLegend, San Diego, CA, USA) were administrated i.p. into MOG_35–55_ peptide immunized mice from the day of disease onset for 3 consecutive days; then, 100 µg anti-mouse IFNAR were administrated i.p. every 4 days. A quantity of 500 µg 3TC (TCI, Tokyo, Japan) or PBS control was administrated by oral gavage into MOG_35–55_ peptide immunized mice from the day of disease onset for 5 days/week. A quantity of 50 mg/kg minocycline hydrochloride (WAKO, Osaka, Japan) or PBS control was administrated i.p. into MOG_35–55_ peptide immunized mice from the day of disease onset for consecutive 2 days; then, 25 mg/kg minocycline was continuously administrated i.p. every 2 days. 

### 2.4. Cell Isolation

Single cell suspensions of splenocytes were generated by mechanical disruption of tissues. To obtain CNS immune cell, the spinal cord was flushed out with PBS, and the brain was removed from the skull. Brain and spinal cord were cut into small pieces, followed by digestion with 1.4 mg/mL Collagenase H and 100 µg/mL DNase I (Roche, Tokyo, Japan) in RPMI for 40 min at 37 °C. The single cell suspension was obtained by filtration through a 70 μm cell strainer and was enriched by a discontinuous 37%/70% percoll gradient centrifugation (GE Healthcare Life Sciences, Tokyo, Japan). To obtain neurons, we first followed the manual of Adult Brain Dissociation Kit (Miltenyi Biotech, Bergisch Gladbach, Germany) by using gentleMACS Octo Dissociator (Miltenyi Biotech). The obtained single cell suspension was then subjected to neuron purification by using a neuron isolation kit with MACS (Miltenyi Biotech).

### 2.5. BMDC Culture

Ly5.1 B6 mice derived bone merrow cells were culture with 20 ng/mL GM-CSF. Half of culture medium was replaced with fresh GM-CSF on day 3 and 6. On day 8, the suspended cells and loosely attached cells were collected. The cells were maturated by 100 ng/mL lipopolysaccharide (LPS) and 20 ng/mL IL-4, for 24 h. CD11c^+^ MHC II ^hi^ F4/80^-^ DCs were sorted by Aria II and then pulsed with indicated antigen for 6–12 h as indicated.

### 2.6. Calcium Flux Assay

The sorted CNS Th cells as responders were loaded with 1 μm Fluo-4 AM and 1 µm Fura-Red AM for 30 min at 37 °C prior to flow cytometric analysis. Both the APCs (BMDC or microglia) and T cells were washed. Intracellular Ca^2+^ elevation was assessed upon engagement of T cells with APC at a ratio of 1:4 following a quick spin down. Mixed cells were kept at 37 °C for 10 min prior to Ca^2+^ flux analysis on a Canto II flow cytometer. Fluo-4 AM and Fura-Red AM emission on Th cells were measured at 488 nm/530 nm and 488 nm/670 nm, respectively. Data were recorded over 5 min. In a pilot experiment, ionomycin stimulation of the CNS Th cells was used as a positive control. Data were analyzed using FlowJo Kinetics Module software V10.7 (Tree Star, Ashland, OR, USA). Using a ratio of Fluo-4 and Fura-Red to reflect the changes in intracellular Ca^2+^ concentration, plots were generated by Prism 8 (GraphPad, San Diego, CA, USA).

In the indicated experiment, Calcium Flux Assay was carried out by Fluo-4 Direct™ Calcium Assay Kit (Thermo Fisher, Waltham, MA, USA). In brief, isolated whole CNS cells or sorted CNS Vβ5.1/5.2 or Vβ8 Th cells were incubated in Fluo-4 Direct calcium reagent loading solution with 5 mM probenecid for 60 min and were then co-incubated with peptide loaded BMDCs at an E:T ratio of 1:4 for 1.5 h. Fluorescence is measured using Promega Glomax for excitation at 490 nm and emission at 510–570 nm. Data are processed using Prism 8.

### 2.7. Flow Cytometry

Cells were treated with anti-mouse CD16/CD32 (BioLegend) to block Fc receptors before staining. Staining was performed in PBS solution containing 5% FCS. For all experiments, dead cells were excluded using an Aqua Live/Dead fixable staining reagent (Invitrogen, Waltham, MA, USA). Monoclonal antibodies CD4 (GK1.5), CD5 (53-7.3), CD8 (53-6.7), CD11b (M1/70), CD11c (N418), CD19 (6D5), CD45 (30F11), CD107a (1D4B), Cx3cr1 (SA011F11), Vβ5.1 (MR9-4), Vβ8 (F23.1), Eomes (Dan11Mag), TCRβ (H57-597), and MHC II I-A/I-E (M5/114.15.2) were obtained either from BD, BioLegend, or eBioscience (San Diego, CA, USA). For surface staining, cells were incubated with monoclonal antibodies for 30 min on ice. In the cases of Eomes intracellular staining for CNS cells, fresh isolated cells were used without re-stimulation. Flow cytometric analysis was carried out on using a FACS Canto II (BD) with a FACS Diva software, and data were analyzed using a FlowJo (Tree Star, Waltham, MA, USA) software V10.7. 

### 2.8. Cell Sorting

From splenocytes, T cells and B cells were isolated using a CD4 T cell MACS isolation kit, respectively, with an AutoMACS separator according to the manufacturer’s instructions (Miltenyi Biotech, Bergisch Gladbach, Germany). For CNS cells, CD45^hi^ CD19^-^ MHC II^+^ CD11c^+^ CD317^hi^ B220^+^ pDC, CD45^hi^ TCRβ^+^ CD4^+^ T cells, and CD45^int^ CD11b^+^ Cx3cr1^+^ microglia cells were sorted using a FACS Aria IIu (BD).

### 2.9. Mouse Vβ TCR Analysis 

Groups of NR4A2 cKO and control mice were immunized with MOG_35–55_ to induce EAE. During the late phase of EAE (day 26–28), single cell suspensions were prepared from CNS tissue. Cells were stained with antibodies against TCRβ and CD4 combined with the Mouse Vβ TCR screen panel kit (BD Biosciences), before fixation and permeabilization with an intracellular transcription factor staining reagent kit (eBioscience). Subsequently, intracellular staining with antibody against Eomes was conducted for flow cytometer analysis.

### 2.10. Real Time qPCR

Total RNA was extracted from cell populations using an RNeasy Mini kit (Qiagen, Germantown, MD, USA) and was then transcribed into cDNA using a first-strand cDNA Kit (Takara, Osaka, Japan) according to the manufacturer’s instructions. Gene expression was measured by real-time q-RT-PCR analysis using a LightCycler instrument (Roche Diagnostics, Tokyo, Japan) with SYBR Green Master (Roche). Primers of ORF-1 and ORF-2 were synthesized as previous described [31]. The other primers were purchased from Qiagen. Expression levels were normalized to the expression of Rplp0 or β2M. 

### 2.11. Microarray Analysis

Expression microarrays were carried out on FACS sorted CNS microglia using GeneChip Mouse Genome 430 2.0 Arrays (Affymetrix, Tokyo, Japan) prepared using an RNeasy Mini kit (Qiagen) and a GeneChip Hybridization, Wash, and Stain Kit (Affymetrix) according to manufacturer’s instructions. Arrays were washed using a GeneChip Fluidics Station 450 and scanned using a GeneChip Scanner 3000 7G. Array data were compiled using an Affymetrix GCOS software. Differential gene expression analysis was performed using MultiplotStudio of GenePattern. Gene Ontology analysis was firstly performed using PANTHER [32]; the readouts were further analyzed by REVIGO (revigo.irb.hr (accessed on 15 January 2021)) to remove redundant GO terms; the remains were visualized in semantic similarity-based or group-compared scatterplots using ggplot2.

In another setting, all upregulated DEGs of LM as compared with IM were processed by FunSet (http://funset.uno (accessed on 14 April 2021)). Significant enriched GO terms (FDR < 0.01) were classified based on semantic similarity; the terms with the largest average semantic similarity with respect to all terms in the cluster were automatically selected as cluster representatives (medoid terms).

### 2.12. Single-Cell Capture, Imaging, and qPCR Analysis

In brief, sorted microglia from intact or EAE mice were suspended with C1 Suspension Reagent. This cell suspension was then loaded on a (10–17 μm) C1 Single-Cell AutoPrep integrated fluidic circuit (IFC) microfluidic chip designed to capture cells and preamplification. Brightfield (10× magnification) images of each capture site were acquired by a BZ-X710 automated microscope using a BZ-X Viewer software (Keyence, Osaka, Japan). Reagent mixes were prepared according to the Fluidigm protocol including harvest reagent, lysis mix, RT mix, and Preamp mix. Spike 1, 4, and 7 were added to the lysis mix for normalization and quality control purposes. (Fluidigm, San Francisco, CA, USA). The IFC was loaded into the C1 system, running Preamp overnight.

Sso Fast EvaGreen Supermix with Low ROX (Bio-Rad, Hercules, CA, USA) was used for qPCR. The 96.96 Dynamic Array IFC was primed using IFC Controller HX and loaded according to Fluidigm protocol. The qPCR was performed using the high-throughput platform BioMark™ HD System (Fluidigm). Data were collected using the BioMark Data Collection software with GE Fast 96 × 96 PCR + Melt v2 protocol (A thermal mixing protocol of 70 °C for 40 min and 60 °C for 30 s and then a hot start protocol of 95 °C for 60 s, followed by 30 qRT-PCR cycles of 96 °C for 5 s and 60 °C for 20 s. A melting protocol of 60 °C for 3 s followed using a 1 °C increase every 3 s up to 95 °C). Single-cell qPCR data were initially processed by the Fluidigm Real-Time PCR Analysis to generate Ct values.

All subsequent analysis of the gene expression data, including tSNE analysis, hierarchical clustering, and volcano plot, were performed using the SINGuLAR Analysis Toolset 3.0 (Fluidigm) in R platform. 

### 2.13. Immunohistochemistry

In brief, cryosections were collected using a Cryostar NX70 cryostat (Thermo Fisher) and fixed in 4% PFA at room temperature. Next, sections were processed by microwave and defatted in 0.2% Triton. Then, sections were blocked in 3% BSA at room temperature. Primary antibody incubations were carried out overnight at 4 °C. The following antibodies were used: rabbit anti-L1 ORF1p (polyclone, Novusbio, Englewood, CO, USA) and Alexa Fluor 488 rabbit anti-NeuN (clone EPR12763, Abcam, Boston, MA, USA). Alexa Flour 647-conjugated AffiniPure Mouse Anti-Rabbit IgG (Jackson ImmunoResearch, West Grove, PA, USA) was used for ORF-1 antibody detection. Sections without primary antibodies were processed in parallel. Sections were mounted with Fluoromount (Southern Biotech, Birmingham, AL, USA) and imaged with a BZ-X710 automated microscope using a BZ-X Viewer software (Keyence).

### 2.14. In Vitro Translation Protein Expression

Isolated total RNA from sorted cell types were in vitro translated into proteins by using Rabbit Reticulocyte Lysate System (Promega, Madison, WI, USA). Expressed proteins were applied for Ca^2+^ flux experiments. Regarding to ORF-1 expression, full-length of ORF-1 was inserted the expression plasmid pRSET vector (Invitrogen) as previously described [33,34]. ORF-1 protein was translated in vitro by using TNT T7 Quick Coupled Transcription/Translation System (Promega). Quality of expressed protein is determined by using TranscendTM tRNA (Promega). Protein size is confirmed on an SDS-PAGE gel.

### 2.15. ORF-1 Peptide Library

An ORF-1 peptide library containing 72 overlapping peptides (length: 15 amino acids, offset: 5 amino acids, purity 70%, 1–3 mg/peptide, Appendix A) and spanning the entire ORF-1 molecule was synthesized (Eurofins, Yokohama, Japan). Peptides were dissolved in dimethyl sulfoxide (Sigma, Kanagawa, Japan).

### 2.16. Statistical Analysis

Statistical analysis was performed using a Prism 8 Software (GraphPad, San Diego, CA, USA) by unpaired Student’s *t* test, one-way or two-way ANOVA as specified, with Tukey’s or Dunnett’s multiple comparisons. A *p* value < 0.05 was considered as significant difference. ns, not significant; *, *p* < 0.05; **, *p* < 0.01; ***, *p* < 0.001; ****, *p* < 0.0001.

## 3. Results

### 3.1. Inhibition of Microglia Activation Attenuates Clinical EAE

We have previously shown that IFN-I has a capacity to drive the development of neurotoxic Eomes^+^ Th cell that appear in the neurodegenerative phase of EAE (“late EAE” hereafter) [35]. Besides DAM, the late, chronic phase of neurodegeneration in a mouse model of AD was also characterized by CNS accumulation of microglia with IFN-I/MHC-II signatures [18], which may precede the final DAM differentiation. We have recently demonstrated that Eomes^+^ Th cells infiltrated into the CNS of mouse models of amyotrophic lateral sclerosis (ALS) and Alzheimer’s disease (AD) secrete neurotoxic granzyme B after encounter with ectopically expressed ORF1 antigen encoded by L1 retrotransposon [25]. As a massive accumulation of cytoplasmic nucleic acids such as L1 DNA/RNA induces IFN-I production and upregulates surface MHC-II that might present ORF1 antigens, these observations prompted us to investigate whether microglia subsets with IFN-I/MHC-II signatures act as an intrinsic hub connecting CNS inflammation to immune-mediated neuronal cell death during the neurodegenerative processes in late EAE.

To evaluate any requirement for microglia in late EAE pathogenicity, we first block microglia activation. Minocycline, a microglial activation inhibitor [36,37], abolished clinical symptoms of late EAE (Figure 1A), with reduced infiltration of CD4^+^ and CD8^+^ T cells into the CNS (Figure 1B,C). In particularly, the proportion of Eomes^+^ Th cells in the CNS was significantly reduced after minocycline treatment (Figure 1D). Although the inhibition of microglial activation did not alter numbers of microglia in the CNS (Figure 1E), the expression of IFN-I genes by microglia was significantly decreased (Figure 1F). These results suggest that activated microglia under chronic neuroinflammation play an important role in driving pathogenic Eomes^+^ Th cell development which cause CNS damage and resulting clinical signs in late EAE.

### 3.2. Differential Gene Expression in Late EAE Microglia

We divided EAE into two stages, early (E) and late (L), based on clinical course (Figure 2A). Microglia isolated from CNS at early stage (EM) or late stage (LM) were profiled for compressive gene expression and compared with microglia from naïve mice (IM). IFN-I signature was modest in microglia from early EAE (EM) but in in late microglia (LM) showed an apparent upregulation of IFN-I related gene expression (Figure 2B,C and Appendix A). A STRING networks analysis from these data highlights the functional enrichment of interacting genes in IFN-I-linked pathways (Figure 2D). Furthermore, we investigated the kinetics of EAE-associated IFN-I expression by microglia. Representative IFN-Is (IFN-α2 and IFN-β1) were increased in microglia during the late stage (Figure 2E,F). The relevance of IFN-I production by microglia in late EAE was confirmed by comparing IFN-I expression level to that professional IFN-I producers, CNS-derived plasmacytoid dendritic cells (Appendix A). Furthermore, Gene Ontology (GO) enrichment analysis indicated that enrichment of PANTHER GO terms including “immune system” and “response to stimuli” is more enriched differentially expressed genes (DEGs) and more statistically significant in LM, suggesting a temporal activation of microglia during late EAE (Appendix A).

### 3.3. Functional Heterogeneity of EAE-Associated Microglia 

As microglia are suggested to be heterogenous by function and phenotype during neurodegeneration, we performed single cell analysis to further investigate microglial diversity in late EAE. Analysis of obtained gene sets using t-SNE dimensional reduction shows that whilst IM and EM cluster together, LM form a distinct separate cluster by on gene expression (Figure 3A). This differential gene expression suggests that LM acquires entirely different characteristics from those in earlier phases of EAE. Individual inflammatory gene modules, such as *H2-Eb1*, *Spp1*, *Apoe*, and *C1qa*, are clearly enhanced in LM as well as IFN-I-related gene modules, such as *Lgals3bp*, *Gbp2*, and *Clec7a* (Figure 3B). In contrast, the expression level of microglial core genes such as *P2ry12* and *Fcrls* was significantly downregulated in LM compared with IM and EM. The C-type lectin receptor *Clec7a* (also known as Dectin-1, CD369) is one of the most differentially expressed genes in LM, with significant upregulation by microglia in late disease versus in early disease or unimmunized mice (Figure 3C–E). Clec7a expression is previously reported as a feature of DAM during Alzheimer disease [17,18], and in our studies the kinetics *Clec7a* expression by microglia is well correlated with the clinical course of EAE (Appendix A). As Dectin-1-Syk-IRF5 signaling induces IFN-I production in dendritic cells [38], we compared the gene expression profile of microglia with high *Clec7a*-expression (Clec7a-hi) and those with low *Clec7a* expression (Clec7a-lo). Although global gene expression is similar between Clec7a-hi and Clec7a-lo microglia in early EAE (Appendix A), Clec7a-hi microglia from late EAE mice represented a more activated and pathogenic phenotype based on expression of key inflammatory genes (Figure 3F), including complement-related, IFN-I-related, and ApoE-Trem2 signaling genes (Appendix A). Next, we sorted microglia from naïve mice by flow cytometry based on Clec7a levels and stimulated with the Clec7a-binding glucan, zymosan. Stimulation led to a higher expression of *IFN-α2/β1* in Clec7a-hi IM compared with Clec7a-lo microglia, as well as increased expression of IFN-I-related genes such as *Gbp2* and *Rtp4* (Appendix A). Zymosan also promoted antigen presenting properties (*H2-Eb1* and *MHC II*) in Clec7a-hi but not in Clec7a-lo microglia (Appendix A). Therefore, Clec7a expression is a good marker for identifying inflammation-primed microglia with IFN-I or MHC-II signature.

### 3.4. IFN-I Is Critical for Induction of Eomes^+^ Th Cells and Development of Late EAE

As late disease was associated with microglia with an IFN-I signature and microglia were critical for full late disease, we wished to the importance of IFN-I signals for pathogenicity in late disease. However, it has been reported that IFN-I has anti-inflammatory properties in the acute phase of EAE [39,40], which raises this mechanism as a potential confounding factor when considering mechanisms in late disease. Therefore, we tested the effect of IFN-I blockade immediately after EAE onset. Anti-IFNAR mAb treatment significantly ameliorated late EAE symptoms without affecting early disease course (Figure 4A). Interestingly, IFNAR treatment also inhibited the accumulation of Eomes^+^ Th cells in the CNS (Figure 4B). Similarly, IFNAR KO mice showed attenuated clinical symptoms especially in the chronic stage of EAE, with reduced Eomes^+^ Th cell infiltrating the CNS (Figure 4C,D). These data suggest that IFN-I may have a direct effect on driving pathogenic Eomes^+^ Th cells in the CNS.

### 3.5. CNS Antigens Act as Trigger for Pathogenesis in Late EAE Disease

Previously we have reported that Eomes^+^ CNS Th cells may be stimulated locally to damage neurons via production of granzyme B [23]; we also observe that granzyme B-deficient mice are protected from late EAE (Appendix A). In addition, Eomes^+^ Th cells are also observed to accumulate in the CNS during neurodegenerative diseases in mouse models and secreted granzyme B following stimulation by putative CNS antigens [25]. This raises the tantalizing possibility that microglia may act as antigen presenting cells to stimulate Eomes^+^ Th cell leading to local activation and downstream CNS inflammatory damage. Functional enrichment gene ontogeny analysis (FunSet) indicated an increase in the microglia capacity to uptake, process, and present antigen during late EAE (Appendix A), and MHC II levels expressed by microglia were increased over the course of disease (Figure 5A). 

As late disease microglia could act as potential antigen presenting cells, we carried out further investigation into putative CNS antigens that may drive Eomes^+^ Th cell activation. Protein derived from not only neurons, but also from LM, was able to active CNS Th cells using bone marrow-derived dendritic cells (BMDC) as antigen presenting cells (Appendix A).

Such assays may be influenced by cell intrinsic antigen presentation factors despite using model antigen presenting cells in the form of BMDC. Therefore, we generated CNS antigens using an in vitro translation (IVT) system with RNA isolated from LM. Identical BMDC were pulsed with the resulting CNS antigens which were used in turn to stimulate CNS-derived Th cells. CNS Th cells were preloaded with a Ca indicator that allowed the visualization of early T cell activation by monitoring Ca^2+^ flux. Both LM and neuron-derived IVT products yielded a Ca^2+^ flux response (Figure 5B). This stimulation was MHC II-dependent as shown its attenuation by MHC II blockade (Figure 5C).

As the CNS Th cell responses to LM-derived IVT products were similar to neuron-derived IVT products, we investigated the possibility that these findings could result from products that LM had previously collected from neurons. Microglia are known to phagocytose CNS debris including dying neurons via a Trem2-dependent manner [17]. Thus, microglia derived from either Trem2KO or control mice were used to stimulate CNS Th cells from late EAE that had been preloaded with a Ca indicator to monitor immediate T cell activation. Late disease microglia induced Ca^2+^ flux, indicating Th cell activation (Figure 5D), whereas this ability of LM to stimulate CNS Th cells was diminished in the absence of TREM2.

Taken together, these data suggest that microglia incorporate putative CNS antigens via Trem2-dependent manner and present them to CNS-derived Th cells in an MHC II-dependent manner.

### 3.6. ORF-1, an Encoded Protein of L1 Retrotransposon May Contribute to CNS Th Cell Activation

Some oligoclonality of T cell receptor (TcR) repertoire in Eomes^+^ Th cells is indicated by Vβ repertoire skewing [35], implying the presence of putative predominant antigen(s). L1 has been previously highlighted as a possible candidate for endogenous antigens under neurodegeneration [41]. We have also recently reported that Eomes^+^ CNS Th cells in a mouse model of neurodegenerative diseases may be responsive to ORF1 antigen [25].

Therefore, we hypothesized that ORF1 protein could directly activate CNS Th cell or not. Firstly, we confirmed that L1 ORF1 protein, a product of L1 gene (Figure 6A), is expressed in the olfactory bulb, hippocampus, subventricular zone, and cortex layers of the brain during EAE (Appendix A). Next, we tested if L1 ORF1 protein could stimulate CNS Th cells. ORF1-pulsed BMDC induced a clear oscillation of Ca^2+^ flux in CNS Th cells, which was abolished in the presence of anti-MHC II blocking mAb (Figure 6B). Interestingly, kinetic analysis revealed that Th cells at early EAE respond poorly to ORF1 peptide, compared with strong responses by late CNS Th cells (Appendix A). These data suggest that a set of ORF1-recognizing Th cells develop in the CNS during transition toward late EAE. To scrutinize possible cellular responses of CNS Th cells against ORF1 protein, we performed an epitope mapping analysis using a 15-mer synthetic peptide library, covering the whole open reading frame of ORF1 with a five amino acid overlap between adjacent peptides (Appendix A). CNS Th cell activation as indicated by Ca^2+^ flux was diverse across individual peptide with several islands of antigenicity observed suggesting that multiple instances of ORF1 protein epitopes driving Th cell activation may occur (Figure 6C), although this would be dependent on the physiologic antigen processing. Collectively, at least a part of CNS Th cells in late EAE mice were recognized and were stimulated by L1-derived ORF1 protein, with a great range of potential antigens across the whole protein.

### 3.7. Blockade of L1 Activation Ameliorates Late EAE

To further explore the significance of L1 activation in the pathogenic process of late EAE, we treated mice with lamivudine (3TC), a non-toxic reverse transcriptase inhibitor that effectively blocks the activity of L1 retrotransposase [42]. Although 3TC treatment had no impact on the onset of acute EAE, late EAE was significantly attenuated and with increased recovery from disease symptoms (Figure 7A). Accordingly, 3TC treatment also suppressed the accumulation of Eomes^+^ Th cells (Figure 7B), and those cells found in the CNS had reduced pathogenic potential as measure by surface CD107a^+^ recruitment. A significant reduction in ORF-1 expression by neuron and microglia was also observed (Figure 7C). Eomes^+^ Th cells in the CNS at late EAE have a somewhat oligoclonal TcR repertoire with increased proportions of particular Vβ usage such as Vβ5.1/5.2 [35]. The 3TC treatment prevented the skewing of the CNS Th cell repertoire, including the Vβ5.1/5.2 subset (Appendix A). This alteration in oligoclonality indicators is even more apparent in the Eomes^+^ Th cell subset, with the dominance of Vβ3, Vβ4, Vβ5.1/5.2, Vβ7, and Vβ11 chains significantly reduced after 3TC treatment (Figure 7D). Finally, we examined the cellular response of Vβ5 and Vβ8 Th cells against pooled peptide mixtures derived from the ORF1 sequence. As shown in Figure 7E, differential T cell responses were observed against different peptide pools, suggesting a diverse range of CNS Th cells recognizing ORF1 proteins. Taken together, these results suggest that ORF1 protein a potent antigen recognized by a broad range of CNS Th cells, including the Eomes-expressing Th subset, and this endogenous antigen can drive local pathogenic Th cells responses leading to neuroinflammation.

## 4. Discussion

In this study, we reveal microglia play pivotal roles in triggering cytotoxic Th cell-mediated pathogenesis of neurodegeneration in late EAE via augmented production of IFN-I and MHC II-dependent presentation of endogenous antigens. These processes are generated by two microglial subsets with IFN-I-induced genes or high expression of MHC II that develop under conditions of chronic neuroinflammation. Furthermore, we revealed that retrotransposon L1 ectopically activated in the inflamed CNS may play a crucial role both as a potent IFN-I inducer through their replication intermediate and as a source of endogenous antigens recognized by CNS Th cells through its encoded protein, ORF1. Inhibition of microglial activation by minocycline or IFN-I signal blockade resulted in an amelioration of clinical sign with fewer Eomes^+^ Th cells infiltrating CNS. Thus, primed microglia are a pathogenic hub of neurodegeneration in late EAE.

Although RRMS is a demyelinating disease caused by autoimmune-mediated oligodendroglial damage, the clinical symptoms of neurodegenerative complication in SPMS are indistinguishable from those observed in neurodegenerative diseases [21]. Due to the low number and sparse distribution of immune cells in the CNS [43], there has been limited evidence for the functional relevance of acquired immunity in neurodegenerative diseases. Our current data suggest that immune-mediated pathology is involved in a number of neurodegenerative CNS disorders. In support of the above argument, we have revealed that the frequency of peripheral Eomes^+^ Th cells (Eomes frequency) acted as a reliable biomarker for SPMS patients at risk of worsening disease (i.e., neurodegeneration) [24]. Accordingly, we have demonstrated that Eomes^+^ Th cells infiltrated the CNS during disease in mouse models of amyotrophic lateral sclerosis (ALS) and Alzheimer’s disease (AD) where they can secrete neurotoxic granzyme B after encounter with ectopically expressed ORF1 antigen encoded by L1 retrotransposon [25].

Neurodegeneration has long been considered as a consequence of neuron-intrinsic homeostatic failure with cytosolic accumulation of noxious protein aggregate. However, a growing body of evidence points the critical roles of glial cells as accomplices to the development of neurodegenerative diseases [8]. In addition, single cell-RNA sequencing and mass cytometry analysis revealed functional heterogeneity of microglia during neurodegeneration [19,20]. In the brain of a mouse model of Alzheimer’s disease, DAM cells were marked by their overexpression of IFN response genes and components of the MHC II pathway [18]. In this study, microglial upregulation of IFN-I signatures and MHC II are shown to be crucial for the development and activation of neurotoxic Eomes^+^ Th cells. As progressive neuron loss leading to motor dysfunction and cognitive impairment are common characteristics of SPMS, the disease-associated microglia may widely bridge neuroinflammation and neurodegeneration via Eomes^+^ Th cells.

Further single cell analysis revealed the heterogeneity of LM in two gene sets, inflammation and IFN-I responses, suggesting that dynamic and diverse alteration of microglial phenotype contributes to entangled pathology of neurodegeneration. Interestingly, DAM-related genes are enriched in Clec7a^+^ microglia, with the Clec7a ligand zymosan inducing microglial IFN-I production. Galectin-9, an endogenous Clec7a ligand and a biomarker for the interferon signature in SLE [44], may be involved in induction of IFN-I under pathologic conditions [45]. Overall, the parallel action of neuroinflammation and neurodegeneration leading to the complex pathology in late EAE could be understood by the activation profiles of LM.

Although IFN-I has been clinically prescribed for preventing RRMS relapses, prolonged neuroinflammatory features of IFN-I are observed in Aicardi–Goutières syndrome (AGS) [27], a rare hereditary disease with a constitutive overexpression of IFN-I due to a monogenic mutation in IFN signal-related genes. However, the neuroinflammatory function of IFN-I has not been previously indicated in a wide variety of neurodegenerative diseases. A recent study suggests that IFN-I response drives neuroinflammation and synapse loss in the Alzheimer disease model [46], implying a neurodegeneration-promoting property of this cytokine. We demonstrated that the attenuation of late EAE together with a clear reduction in the CNS accumulation of Eomes^+^ Th cells after IFN signal blockade, suggesting that prolonged IFN-I expression is harmful for CNS homeostasis.

We demonstrate that CNS Th cells respond to neuron- or microglia-derived antigens but not to those expressed in splenocytes. Intriguingly, Trem2-deficient microglia are blunted to activate CNS Th cells, and late EAE-derived microglia directly induce Ca^2+^ influx responses in CNS Th cells, suggesting that both neuron and microglia act as sources of the antigens. Presentation of endogenous antigens to MHC II in microglia may embed autophagy for recruitment of proteins to MIIC compartment, in which IFN-I-induced autophagy may be involved in [47]. Although antigen presenting capacity of microglia is not necessarily well characterized, our data clearly indicate that microglia acquire MHC II expression upon stimulation and become efficient antigen presenters under pathogenic conditions with neurodegeneration. Noncoding regions are shown to be the main source of targetable tumor-specific antigens due to the alteration of epigenetic landscape under chronic inflammation [48], and aberrantly expressed endogenous retroelements draw special attention [49], because a different set of retroelement aberrantly expressed under chronic inflammation provides potentially immunogenic neoantigens [50]. Accordingly, skewed TCR Vβ repertoire between Eomes^+^ and Eomes^-^ Th [35] prompted us to explore predominant antigen(s) recognized by Eomes^+^ T cells. In this regard, massive and continuous production of IFN-I without an apparent trace of microbial infection commonly observed in SLE [51], interferonopathy AGS [52], ALS [53], and aging [54] hinted at a dysregulated activation of L1. There are about half million copies of L1 gene that occupy ~18% of whole human genome, and 100 copies and 3000 copies of them are retrotransposition-competent in the human and mouse genome, respectively. L1 is strongly suppressed in most somatic cells under physiological conditions via epigenetic and non-epigenetic mechanisms. Indeed, L1-induced IFN-I production in glial cells is required to counteract on propagation of L1 [55]. Under chronic inflammation, epigenetic dysregulation allows ectopic L1 expression encoding the ORF1/ORF2 protein, and the RNP complexes translocate to the nucleus. Then, ORF2 with endonuclease and reverse transcriptase activity induces target-primed retrotransposition in a copy-and-paste manner. Although the detrimental effect of L1 retrotransposon has been exclusively highlighted for disruption of functional genes or dysregulated expression of neighboring genes after germline insertions, the current study demonstrated that nucleic acid intermediates and protein product of the L1 may have additional immune-mediated roles that promote neurodegeneration. In fact, the ORF1 protein expressed in either neuron or microglia is able to stimulate CNS Th cells. L1 overexpression induces hyper IFN-I production via detection of nucleic acid intermediates after RIG-I/MDA5-mediated RNA sensing or cGAS/STING-mediated DNA sensing [56]. Reverse transcriptase inhibitor 3TC that inhibits L1 retrotransposition attenuates late EAE disease with a suppression of Eomes^+^ Th cells in the CNS, suggesting that ectopic L1 expression promotes Eomes^+^ Th cells-dependent neurodegeneration. Furthermore, in vitro-translated ORF1 protein enhanced TcR-mediated Ca^2+^ influx in CNS Th cells. It is not clear whether host immunity establish tolerance against ORF1 or not; antibodies against ORF1 were detectable in a population of SLE patients with severe and active disease [57], suggesting ORF1 may be recognized as an antigen under chronic inflammation. To our knowledge, this is the first report describing retrotransposition-independent and immune response-mediated detrimental action of L1 in neurodegeneration. Cellular response of CNS Th cells to ORF1 peptides library revealed that the overall Th cell-stimulating property was gradually increased over the course of disease progression. Similarly, Vβ repertoire skewing in CNS Th cells, especially that usually manifest in Eomes^+^ Th cells [35], was diminished after 3TC treatment, further supporting a link between ORF-1 and CNS Th cells. As the abolished Vβ repertoire was not limited to Vβ5.1/5.2, further study will unveil a more detailed mechanisms of ORF1-mediated priming of CNS Th cells.

This current study demonstrated that chronic inflammation provoked a functional fluctuation in microglia and Eomes^+^ Th cells in the CNS, mutual exchange of which forms a previously unappreciated vicious network that leads to neurodegeneration through upregulation of IFN-I and antigen presentation of CNS proteins (Appendix A). To sum up, our data highlight a critical contribution of microglia with IFN-I or MHC II signature as a neuroinflammatory hub that is essential for immune-cell mediated neurodegeneration.

## Figures and Tables

**Figure 1 cells-12-00868-f001:**
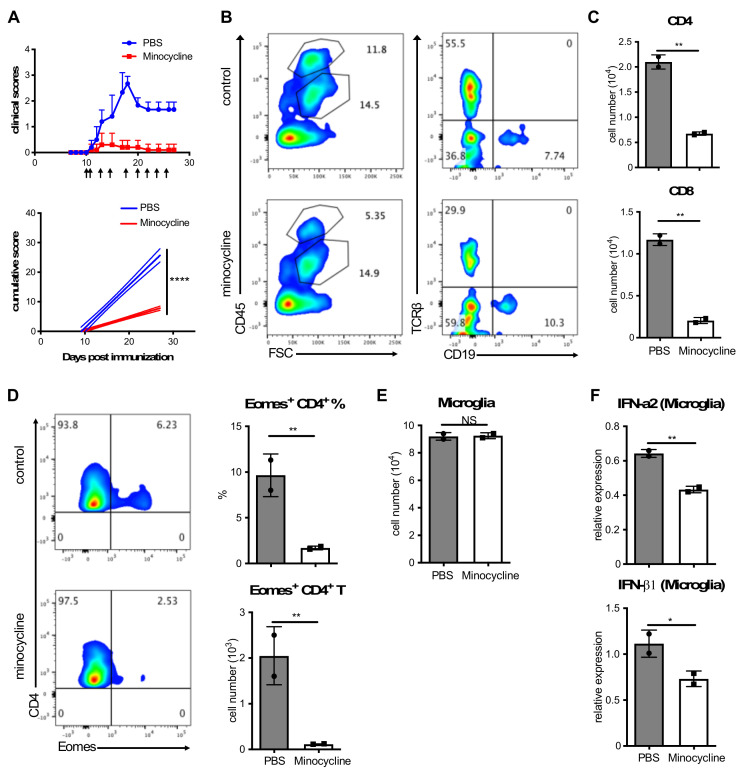
Interfering activation of microglia attenuates EAE disease. (**A**) EAE-induced CD4-Cre Nr4a2 cKO mice were treated with i.p. injections of minocycline or PBS control at indicated time points (See also Method section). One representative experiment out of two is shown. Clinical EAE scores are shown via error bars representing SEM. In the bottom panel, solid lines represent cumulative disease scores; dashed lines indicate the 95% confidence intervals; linear regression analysis; ****, *p* < 0.0001. (**B**–**D**) Freshly isolated CNS cells were stained and detected by a flow cytometry. Flow cytometric plots show representative data. The cell number was shown using a bar plot (**C**). Error bars represent the mean ± SD values; Student’s *t*-test; **, *p* < 0.01. (**D**) Eomes Th cells were analyzed by FACS. The cell number was shown using a bar plot (right). Error bars represent the mean ± SD values; Student’s *t*-test; **, *p* < 0.01. (**E**) Data for the cell number of microglia were shown as a bar graph. Error bars represent the mean ± SD values; Student’s *t*-test; NS, no significant differences. (**F**) The expression levels of IFN-I in microglia were determined by qPCR. Error bars represent the mean ± SD values; Student’s *t*-test; *, *p* < 0.05; **, *p* < 0.01.

**Figure 2 cells-12-00868-f002:**
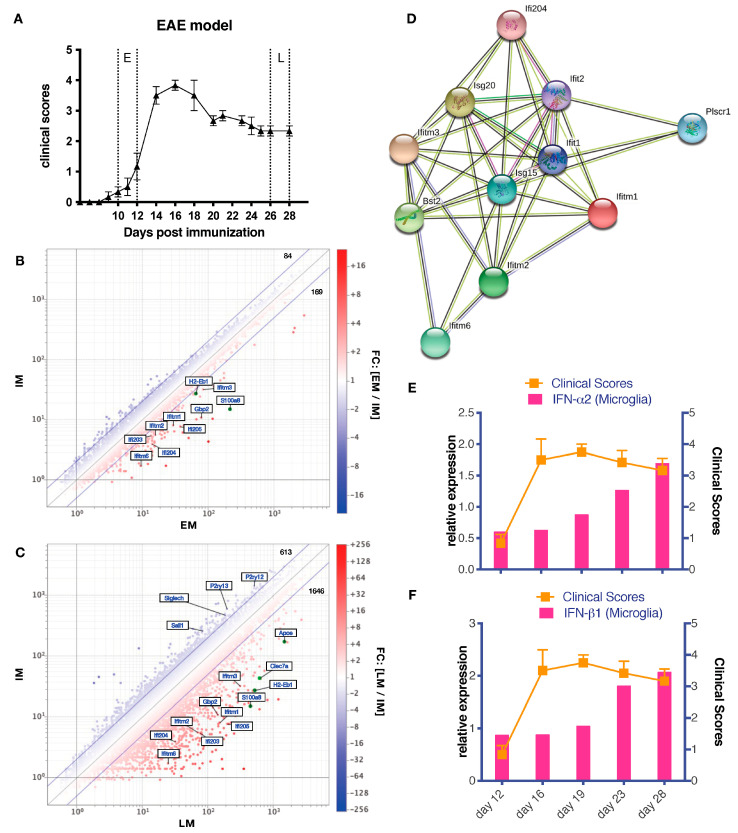
Differential gene expression in late EAE microglia. (**A**) A diagram of the time points in EAE mouse model. E, early EAE, day 10–12; M, peak or mid EAE, day 16–18; L, late EAE, day 26–28 post immunization. (**B**,**C**) Gene expression in microglia. At different time points (intact, early, and late), Cx3cr1^+^ CD45 ^int^ microglia were isolated from CNS of intact or EAE Cx3cr1−GFP mice for expression microarray analysis. Two different time point samples were compared and analyzed using a scatter plot. DEGs were highlighted by fold change ≥2. The genes that were related with this study were labeled. IM, intact microglia; EM, early EAE microglia; LM, late EAE microglia. (**D**) Interaction of enriched GO “type I interferon response” genes are shown by STRING. (**E**,**F**) The bar graph of the kinetics of IFN−I expression in microglia detected by qPCR is combined with the line graph of clinical scores. Experiment was repeat performed three times; representative data of IFN−I expression in microglia are shown.

**Figure 3 cells-12-00868-f003:**
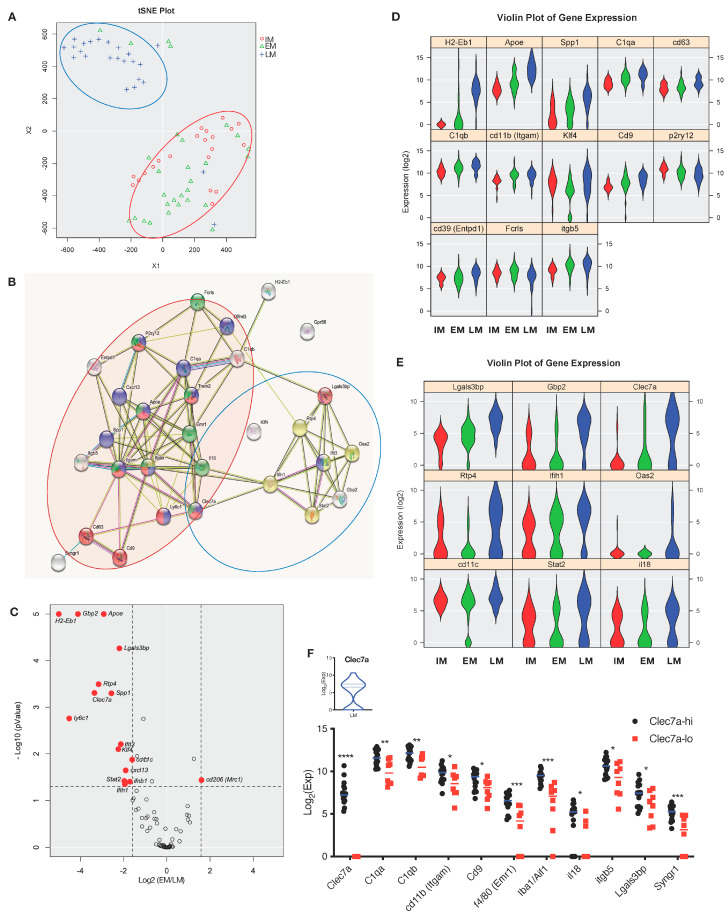
Gene expression profile of microglia associated with EAE pathogenesis. Single−cell analysis of microglia. Isolated microglia from the whole brain of intact and EAE Cx3cr1−GFP mice. IM, intact; EM, early EAE (Day 10–12 post immunization); LM, late EAE (Day 26–28 post immunization) (**A**) tSNE plot of intact and microglia from EAE. (**B**) Gene sets of detected genes. Red circle, activation markers; blue circle, IFN−I response genes. (**C**) Volcano plots for intact and EAE microglia. Average log 2 (fold change) versus−log 10 (*p*−value) for all genes. (**D**) The expression level of inflammatory panel or (**E**) IFN−I related genes was shown as the violin plots. (**F**) Microglia from late EAE were divided into Clec7a ^hi^ and Clec7a ^lo^ subsets. Distribution of Clec7a expression in LM is shown as a violin plot. Expression of detected genes are shown by a bar graph. Error bars represent the mean ± SD values. Student’s *t*−test; *, *p* < 0.05; **, *p* < 0.01; ***, *p* < 0.001; ****, *p* < 0.0001.

**Figure 4 cells-12-00868-f004:**
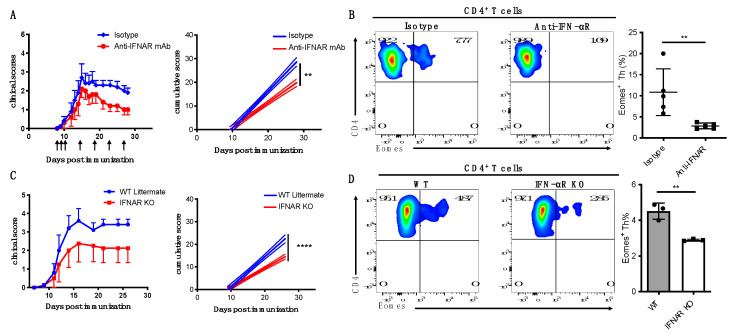
Blocking IFN−I signals brings clinical benefits. (**A**,**B**) EAE-induced WT B6 mice were treated with i.p. injections of anti−IFN−αR antibodies or control IgG at indicated time points. (**A**) Clinical EAE scores are shown via error bars representing SEM. In the right panel, solid lines represent cumulative disease scores; dashed lines indicate the 95% confidence intervals; linear regression analysis, **, *p* < 0.01 (**B**) Freshly isolated CNS CD4^+^ T cells were intracellularly stained for Eomes. Flow cytometric plots show representative data. Summary data were shown using a scatter plot (right). Error bars represent the mean ± SD values. Student’s *t*−test; **, *p* < 0.01. (**C**,**D**) EAE using IFNα/βR KO mice or WT littermates. (**C**) Clinical EAE scores are shown via error bars representing SEM. In the right panel, solid lines represent cumulative disease scores; dashed lines indicate the 95% confidence intervals; linear regression analysis, ****, *p* < 0.0001. (**D**) FACS plots show representative data of Eomes staining (**left**). Summary data of frequency of Eomes^+^ CD4^+^ Th cell (**right**). Error bars represent the mean ± SD values. Student’s *t*−test; **, *p* < 0.01.

**Figure 5 cells-12-00868-f005:**
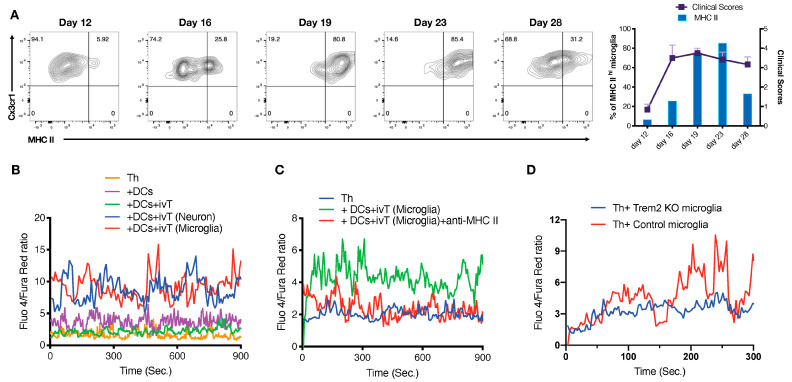
Miscellaneous CNS antigens stimulate CNS-infiltrating Th cells in late EAE. (**A**) The kinetic of MHC II expression in microglia was evaluated by FACS. Represented FACS plots (left) and correlation with clinical score (right) are shown. Experiment was repeat performed three times; representative data are shown. (**B**,**C**) Sorted WT CNS Th cells from peak EAE were stimulated by antigen-loaded BMDCs. Ca^2+^ flux of CNS Th cells was measured by FACS, a liner graph was shown as ratio of Fluo 4 vs. Fura Red. ivT, in vitro translation regent; LNivT, in vitro translated protein of neuron from late phase of EAE (Day 28); LMivT, in vitro translated protein of microglia from late phase of EAE (Day 28) (**B**) BMDCs were pulsed with in vitro translated protein of microglia or neuron from late EAE. (**C**) Sorted WT CNS Th cells from peak EAE were stimulated by antigen−loaded BMDCs in presence or absence of anti−MHC II mAb. (**D**) CNS Th cells and microglia from Trem2 cKO or WT of late EAE were isolated. CNS Th cells were shortly stimulated by microglia. Ca^2+^ flux of CNS Th cells was measured by a FACS Canto II, a liner graph was shown as ratio of Fluo 4 vs. Fura Red.

**Figure 6 cells-12-00868-f006:**
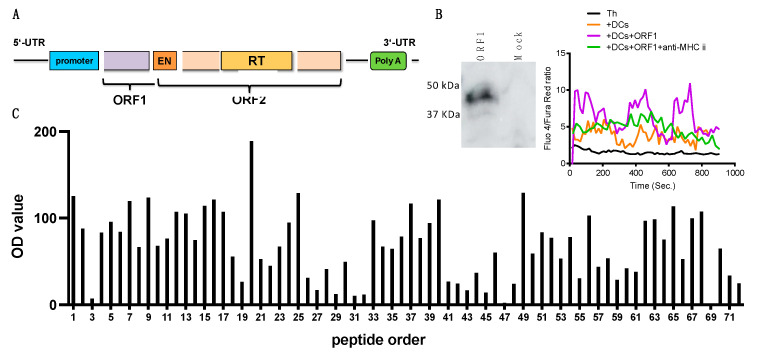
ORF−1, an encoded protein of L1 retrotransposon, may contribute to CNS Th cell activation. (**A**) Genetic structure of murine L1. (**B**) Quality of in vitro translated protein was confirmed (left). 1: ORF1; 2: mock. In the right panel, sorted WT CNS Th cells from peak EAE were stimulated with or without ORF1−loaded BMDCs in presence or absence of anti−MHC II mAb. Ca^2+^ flux of CNS Th cells was measured by FACS, shown as ratio of Fluo 4 vs. Fura Red. (**C**) CNS Th cells were shortly co−cultured with BMDCs that were pulsed with synthesized ORF1 peptide library. Ca^2+^ flux of CNS Th cells was shown as a bar graph determined by OD value. See also the Method section.

**Figure 7 cells-12-00868-f007:**
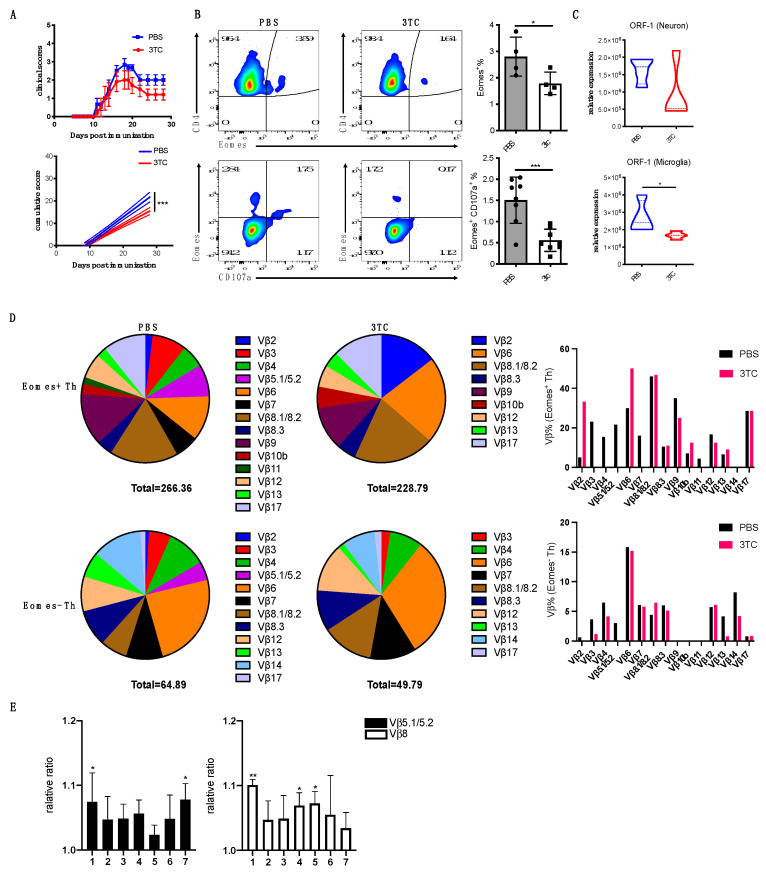
Blocking L1 activity ameliorates late EAE. (**A**) EAE-induced CD4−Cre Nr4a2 cKO B6 mice were treated with 3TC or PBS control (oral garage) as described in Method section. Clinical EAE scores are shown via error bars representing SEM. In the bottom panel, solid lines represent cumulative disease scores; dashed lines indicate the 95% confidence intervals; linear regression analysis. ***, *p* < 0.001. (**B**) Expression level of Eomes and CD107a is measured by FACS. FACS plots (left) show the representative data. Summary data are shown as bar graphs (right). Student’s *t*−test; *, *p* < 0.05; ***, *p* < 0.001. (**C**) Expression of ORF1 in neuron and microglia from control and 3TC-treated EAE mice was shown as bar graphs determined by qPCR. Student’s *t*−test; *, *p* < 0.05. (**D**) Sorted CNS Th cells were surface stained using mouse Vβ TCR screen panel kit. Then intracellular staining was performed against Eomes. Frequency of each Vβ subset in Eomes^+^ and Eomes^−^ CD4^+^ T cells are summarized and shown as pie graphs. See also Appendix A. (**E**) Sorted Vβ5.1/2 and Vβ8 CNS Th cells were shortly co-cultured with BMDCs that were pulsed in presence or absence of three mixed synthesized ORF−1. Ca^2+^ flux of CNS Th cells was shown as a bar graph using the relative ratio of OD510 by comparing with negative control. Error bars represent the mean ± SD values. Student’s *t*-test; *, *p* < 0.05; **, *p* < 0.01.

## Data Availability

All data are available from the corresponding author Shinji Oki upon reasonable request.

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
