# Peer review of "Pathogenic Microglia Orchestrate Neurotoxic Properties of Eomes-Expressing Helper T Cells"

_cells, 2023, doi:10.3390/cells12060868_

Round 1

Reviewer 1 Report

Dear Editor,

The manuscript “Pathogenic microglia orchestrate neurotoxic properties of Eomes-expressing helper T cells” aims to address the contribution of specific subsets of microglia (those presenting IFN-signatures) in regulating neurotoxic properties of Eomes+ T cells in the experimental autoimmune encephalitis (EAE) model of multiple sclerosis. To prove their hypothesis, the authors utilise a series of conditional/inducible mice harbouring targeted gene deletions of either NR4 or TREM2 in microglia and global knockout animals for interferon-alpha and Granzyme B genes.

Overall, the study is well designed and provides convincing mechanistic insights into the role of specific microglia subsets (with IFN-I signatures or elevated MHC-II) in triggering Th cell-mediated neurotoxic responses via L1 antigen during late stage EAE.

However, some issues require the authors’ attention:

Major

1.       No supplementary files are available, although these are referred to in the text in multiple instances.

2.       Methods describe details of neuronal isolation and immunohistochemistry protocols. However, none of the results in the main text seem to show results pertaining to neurons or IHC. Perhaps these results are in the missing supplementary data?

Minor

3.       Typo (Page 2, line 48) “….fluctuations..”

4.       Typo (Page 3, lines 116-117) “….for 3 consecutive days…”

5.       Typo (Page 3, line 122) “..minocycline was continuously administered…”

6.       Typo (Page 4, line 183) “…immunised with MOG35-55 to induce EAE…”

7.       Typo (Page 6, line 270) “…multiple comparisons…”

Reviewer 2 Report

The manuscript “Pathogenic microglia orchestrate neurotoxic properties of 2 Eomes-expressing helper T cells” by Zhang et al., identified microglia with MHC-II and/or IFN-I 12 signatures and highlighted the importance of these subset of microglia in neuroinflammatory diseases. The set of experiments performed are scientifically sound and well represented in the manuscript. The manuscript in present form may be considered for publication. Author’s may incorporate a schematic figure depicting the findings of the manuscript would attract more attention and readers.
